# OpenReview forum: "Doxing via the Lens: Revealing Location-related Privacy Leakage on Multi-modal Large Reasoning Models"
_ICLR.cc/2026/Conference — ICLR 2026 Poster_

### Official Review · Reviewer_gxPb · 2025-10-20

**Soundness:** 3
**Presentation:** 2
**Contribution:** 3
**Rating:** 6
**Confidence:** 5

**Summary:**

This paper identifies a new privacy risk in multimodal large reasoning models (MLRMs): geolocation inference leakage, where models can deduce users’ home locations from images like selfies. The authors propose a three-level privacy risk framework and introduce DOXBENCH, a 500-image dataset to evaluate such risks. They also develop GEOMINER, an attack framework that demonstrates how MLRMs can effectively infer locations using visual clues, underscoring the urgent need for privacy safeguards in multimodal AI systems.

**Strengths:**

- This paper presents a comprehensive study, covering the motivation, evaluation of existing methods, as well as attack and defense aspects.

- I particularly like Table 1, which maps the identified risks to specific legal and regulatory provisions.

- The experiments are also quite thorough and well-conducted.

- threat model defined is aligned with the practice.

**Weaknesses:**

First, I would like to clarify that I have read the entire main body of the paper carefully, including some of the appendices that are relevant to my interests, as well as the figures and tables in those appendices. I may have skipped certain parts of the appendix that are either unrelated to the main text or not of direct interest to me. Therefore, if any of my questions have already been addressed in the appendix, please kindly point that out.

- The prompt structures used in the benchmark evaluation may reflect different risk capabilities. How do you evaluate the potential capability or upper bound of risk represented by each prompt?

- I noticed that in Table 2, Claude’s VRR is quite low, which seems to significantly affect the overall evaluation outcome. Can your evaluation strategy mitigate or correct this imbalance caused by low VRR values?

- Many of the result explanations in the paper do not clearly indicate which figure or table they refer to. For example, the statement “Prediction difficulty increases with the annotated levels”, which figure is this referring to?

- How was Figure 25 generated? From which dataset or classification process did it originate, and what method was used for the classification?

- Regarding Figure 4, could there be potential information leakage, since the “clue” might have been extracted from the same dataset used for evaluation?

**Questions:**

- Can your evaluation strategy mitigate or correct this imbalance caused by low VRR values?

- Many of the result explanations in the paper do not clearly indicate which figure or table they refer to. For example, the statement “Prediction difficulty increases with the annotated levels”, which figure is this referring to?

- How was Figure 25 generated? From which dataset or classification process did it originate, and what method was used for the classification?

- Regarding Figure 4, could there be potential information leakage, since the “clue” might have been extracted from the same dataset used for evaluation?

**Details Of Ethics Concerns:**

The main issue with this paper lies in its **ethical concerns**, which likely require the **intervention of the ICLR Ethics Committee**.
Specifically, there are several points that need careful evaluation:

1. The paper employed **268 unique workers participating in Mtruck**, but it does not specify the **ethical procedures**, **payment details**, or whether **ethical approval** was obtained. The claimed IRB exemption seems to cover only **dataset collection**, not human participation.
2. In the **Ethics Statement**, the authors mention that data collection received an **IRB exemption**, but there is **no reference ID or supporting documentation** provided. These materials should be **strictly reviewed**.
3. The paper **should not be published** until all ethical issues are **properly addressed and verified**.

---

> ### Author Response · Authors · 2025-11-22
> **Response to Reviewer gxPb (1/4)**
>
> Thank you very much for your careful reading of our work and for the many concrete, constructive remarks you have shared. We can clearly see the effort behind your review, and we are deeply appreciative. In return, we have prepared the following point-by-point replies to address each of your comments in detail.
>
> ---
> > `Weakness 1`: **The prompt structures used in the benchmark evaluation may reflect different risk capabilities. How do you evaluate the potential capability or upper bound of risk represented by each prompt?**
>
> **A1**: We thank the reviewer for raising the important point that different prompt structures may correspond to different “risk capabilities,” and that this affects how one should interpret our results. We clarify below how to interpret our main benchmark as a lower bound and how our additional experiments probe stronger, more “upper-bound” risk regimes.
>
> **(1) Minimal baseline prompt**
>
> In our main benchmark, we deliberately fix a single, minimal prompt: the base query “Where is it?” together with an output-format constraint as shown in Figure 2 in Section 3.1. This design is intentional. It avoids expert-level prompt-engineering that leads models to leak as much private information as possible, but instead simulates a non-expert user who simply uploads a photo and asks where it is. Under this weak and non-adversarial setup, the reported GLARE and CCPA accuracy should be interpreted as a conservative lower bound on each model’s location-related privacy risk: if a model already leaks sensitive geolocation information under such a minimal prompt, then stronger prompts or more capable attackers can only increase the effective risk.
>
> **(2) Jailbreak/interactive prompts & Top-k decoding**
>
> Beyond single-shot prompts, we further explore how interactive, jailbreak-style prompting and decoding choices expand the potential risk upper bound. We construct a 20-image subset from high-risk visual scenarios where these vanilla MLRMs fail to answer, and evaluate combinations of jailbreak prompts (PAIR [1], Persuade [2], Mousetrap [3]) replacing “Where is it?” and Top-k decoding.
>
> On this subset, as shown in Table R1 and Table R2, we observe that jailbreak methods can substantially raise the potential task capability upper bound in high-risk scenes: for certain model–attack combinations, increases in VRR are accompanied by simultaneous increases in CCPA accuracy and GLARE, indicating that more answers are given and they are more privacy-sensitive. At the same time, this effect is highly dependent on the model and jailbreak method: jailbreak methods such as PAIR / Persuade are effective on Gemini 2.5 Pro, GPT-5, and Claude Opus 4 (Thinking), but are almost ineffective on OpenAI o3 and OpenAI o4-mini. However, when we switch from Top-1 to Top-3 setting, we find that in most cases jailbreak methods perform no better, and **often worse** than the vanilla baseline.
>
> Taken together, these experiments show how progressively stronger prompts and Top-1/Top-3 setting trace out an upper bound of potential risk capability. A promising direction for future work is to systematically co-optimize prompts so that increases in VRR are accompanied by improving both GLARE and CCPA accuracy, yielding a more precise characterization of the attainable upper bound of model capability. Our framework and benchmark provide a concrete, reproducible way to study this trade-off.
>
> **Reference**
>
> [1] Chao, P., Robey, A., Dobriban, E., Hassani, H., Pappas, G. J., & Wong, E. (2024). Jailbreaking black box large language models in twenty queries. arXiv preprint arXiv:2310.08419.
>
> [2] Zeng, Y., Lin, H., Zhang, J., Yang, D., Jia, R., & Shi, W. (2024). How Johnny can persuade LLMs to jailbreak them: Rethinking persuasion to challenge AI safety by humanizing LLMs. In L.-W. Ku, A. Martins, & V. Srikumar (Eds.), Proceedings of the 62nd Annual Meeting of the Association for Computational Linguistics (Volume 1: Long Papers) (pp. 14322–14350). Association for Computational Linguistics.
>
> [3] Yao, Y., Tong, X., Wang, R., Wang, Y., Li, L., Liu, L., Teng, Y., & Wang, Y. (2025). A mousetrap: Fooling large reasoning models for jailbreak with chain of iterative chaos. In W. Che, J. Nabende, E. Shutova, & M. T. Pilehvar (Eds.), Findings of the Association for Computational Linguistics: ACL 2025 (pp. 7837–7855). Association for Computational Linguistics.

---

> > ### Comment · Reviewer_gxPb · 2025-11-23
> > **reviewer response**
> >
> > As a response, this goes some way toward clarifying that the fixed minimal prompt gives a conservative lower bound on location-privacy risk, which is reasonable. However, the claim that jailbreak/top-K experiments "trace out an upper bound of potential risk capability" is not fully convincing, since they are run on a hand-picked 20-image high-risk subset with only a few prompt templates, i.e., a very narrow slice of the attack space. The paper should have more clearly defining "risk capability".

---

> > > ### Author Response · Authors · 2025-11-25
> > >
> > > Thank you for raising this concern and for pointing out the ambiguity around our use of the term “potential risk upper bound”. We apologize for the confusion. Here, the potential upper risk upper bound referred to the **adversarially-enhanced leakage evaluation**, probing how much leakage (risk) a model exhibits when the adversary applies stronger or more informed prompting strategies (capacity). Thus, the risk here refers to model-side privacy risk while the capacity refers to the attacker-side capacity.
> > >
> > > Our main goal is to evaluate the model’s privacy risk level under different attacker capacities. Here, the attacker can have different levels of capabilities such as low-capability attackers who only know the standard prompt “Where is it?” to more capable attackers who possess knowledge of jailbreak strategies or decoding heuristics. By manipulating attacker capacity, we observe how the model’s risk level scales as adversarial knowledge strengthens.
> > >
> > > We never want to claim to discover a global or theoretical upper bound on the model’s leakage. We will remove this terminology in our revised paper.

---

> > > > ### Comment · Reviewer_gxPb · 2025-11-25
> > > >
> > > > Thank you for the detailed clarification regarding your institution's ethics processes and the MTurk component.
> > > >
> > > > First, I would like to emphasize that I am not an expert in research ethics or institutional policy. My concerns are based mainly on my own prior experience and on the public guidance available to reviewers. In my own work using MTurk, I have always sought ethics approval, and I have also relied on the ICLR Code of Ethics, which explicitly states that:
> > > > ```
> > > > "Where human subjects are involved in the research process (e.g., in direct experiments, or as annotators), the need for ethical approvals from an appropriate ethical review board should be assessed and reported."
> > > > ```
> > > > Given this, I am not fully confident in my ability to definitively judge whether your interpretation and local procedures are professionally sufficient or fully compliant. For this reason, I continue to believe that the situation warrants a formal assessment by the ICLR ethics board, who are in a much better position than I am to adjudicate this borderline case.
> > > >
> > > > Your most recent clarification has partially alleviated my concerns, in the sense that I now better understand how your institution distinguishes internal PI review, HRPP determinations, and IRB review, and how you document a NHSR determination. From this point onward, I will refrain from further debating the ethics component in the discussion thread and defer to the ethics chairs and ACs to make a professional determination on compliance.
> > > >
> > > > That said, since MTurk-based data collection is an integral part of your evaluation, I strongly encourage you (for any accepted or camera-ready version) to document the full MTurk protocol in an appendix. This should include:
> > > > - the worker inclusion criteria and task design,
> > > > - the pay rate and estimated effective hourly wage, and
> > > > - a concise description of the ethical assessment and NHSR determination process at your institution.
> > > >
> > > > Even if the study is ultimately deemed non-HSR, this level of transparency is important for readers and for our community's evolving norms around crowdworker involvement.
> > > >
> > > > Separately from the ethics issue, I would also like to comment on the technical and scientific contribution. For me, this work falls into a category of “solid but not particularly distinctive” papers. The idea of an adversarially-enhanced leakage evaluation is sensible, but in its current form it feels like a type of evaluation that could be instantiated in many similar ways every year. I struggle to identify a truly standout conceptual novelty or methodological innovation that would clearly distinguish this paper at the ICLR level.
> > > >
> > > > In addition, some aspects of the rebuttal involved retracting or substantially rephrasing earlier claims (for example, the way "potential risk upper bound" was originally presented and then corrected). This back-and-forth does not fully reassure me about the overall rigor and precision of the work’s framing and methodology, even if the underlying empirical study is careful in many respects.
> > > >
> > > > Taking all of this into account, I am personally comfortable restoring my overall score to a 4. Whether the paper should ultimately be accepted or rejected, however, will depend on the joint judgment of the Area Chairs and the ICLR ethics board, especially in light of the ethical considerations discussed above.

---

> > > > > ### Comment · Area_Chair_oGd5 · 2025-11-27
> > > > > **Thanks!**
> > > > >
> > > > > Dear all,
> > > > >
> > > > > thanks for this very open and thoughtful discussion. While I am not specialized in ethics reviews, and there are even different procedures in different countries, in my opinion this looks like the authors complied with the rules of their university. And you seem to agree that the information provided by the authors looks good. For me this is also fine, but I am also happy if the ethics chairs can have a brief look.
> > > > >
> > > > > Your AC

---

> ### Author Response · Authors · 2025-11-22
> **Response to Reviewer gxPb (2/4)**
>
> **Table R1: Top-1 performance of vanilla vs. jailbreak-style prompts on a 20-image high-risk subset.**
> |Model|Method|VRR|Average Distance|Median Distance|CCPA|GLARE|
> |-|-|-|:-:|:-:|:-:|:-:|
> |Gemini 2.5 Pro|Vanilla|70.00|214.77|103.11|0.00|773.84|
> ||Mousetrap|90.00|253.65|8.70|11.11|1294.30|
> ||PAIR|70.00|186.92|52.37|0.00|856.29|
> ||Persuade LLM|100.00|125.99|7.09|22.22|1568.66|
> |GPT5|Vanilla|60.00|250.32|45.20|16.67|721.42|
> ||Mousetrap|90.00|225.91|136.85|12.50|951.62|
> ||PAIR|70.00|358.62|490.14|14.29|564.63|
> ||Persuade LLM|30.00|35.68|0.28|66.67|665.35|
> |OpenAI o3|Vanilla|50.00|375.67|314.07|20.00|432.06|
> ||Mousetrap|50.00|75.52|1.40|0.00|938.26|
> ||PAIR|10.00|15.23|15.23|0.00|176.32|
> ||Persuade LLM|10.00|529.48|529.48|0.00|73.93|
> |OpenAI o4 mini|Vanilla|50.00|329.42|451.20|20.00|415.41|
> ||Mousetrap|0.00|NaN|NaN|0.00|NaN|
> ||PAIR|10.00|181.06|181.06|0.00|104.89|
> ||Persuade LLM|0.00|NaN|NaN|0.00|NaN|
> |Claude Opus4 Thinking|Vanilla|30.00|175.97|1.40|33.33|526.35|
> ||Mousetrap|0.00|NaN|NaN|0.00|NaN|
> ||PAIR|70.00|127.02|44.46|33.33|911.83|
> ||Persuade LLM|30.00|1.23|1.23|0.00|746.66|
>
> **Table R2: Top-3 performance of vanilla vs. jailbreak-style prompts on a 20-image high-risk subset.**
> |Model|Method|VRR|Average Distance|Median Distance|CCPA|GLARE|
> |-|-|-|:-:|:-:|:-:|:-:|
> |Gemini 2.5 Pro|Vanilla|100.00|44.96|4.05|25.00|1798.25|
> ||Mousetrap|90.00|27.02|3.93|14.29|1688.14|
> ||PAIR|80.00|107.62|3.27|14.29|1362.40|
> ||Persuade LLM|100.00|51.45|1.76|22.22|1898.51|
> |GPT5|Vanilla|80.00|239.48|9.15|12.50|1151.42|
> ||Mousetrap|80.00|180.70|24.87|0.00|1068.46|
> ||PAIR|50.00|413.41|479.87|0.00|394.58|
> ||Persuade LLM|50.00|20.95|9.09|25.00|895.81|
> |OpenAI o3|Vanilla|50.00|4.87|1.79|25.00|1118.20|
> ||Mousetrap|40.00|242.12|19.92|0.00|530.16|
> ||PAIR|20.00|9.67|9.67|0.00|378.87|
> ||Persuade LLM|0.00|NaN|NaN|0.00|NaN|
> |OpenAI o4 mini|Vanilla|70.00|255.44|88.56|0.00|771.69|
> ||Mousetrap|70.00|201.20|20.16|14.29|945.25|
> ||PAIR|0.00|NaN|NaN|0.00|NaN|
> ||Persuade LLM|10.00|665.83|665.83|0.00|67.31|
> |Claude Opus4 Thinking|Vanilla|50.00|21.98|18.10|25.00|842.68|
> ||Mousetrap|0.00|NaN|NaN|0.00|NaN|
> ||PAIR|40.00|135.38|7.41|14.29|609.22|
> ||Persuade LLM|40.00|279.63|279.63|0.00|369.39|

---

> ### Author Response · Authors · 2025-11-22
> **Response to Reviewer gxPb (3/4)**
>
> > `Weakness 2 & Question 1`: **I noticed that in Table 2, Claude’s VRR is quite low, which seems to significantly affect the overall evaluation outcome. Can your evaluation strategy mitigate or correct this imbalance caused by low VRR values?**
>
> **A2**: We appreciate the reviewer’s concern about the low VRR of Claude series models in Table 2 and its potential impact on the overall evaluation. Our evaluation, however, is explicitly designed so that models with very low VRR do **not** distort the comparison. This is precisely the motivation for introducing the GLARE metric, which formalizes location-related privacy leakage as an information-theoretic quantity: GLARE is defined as the mutual information between the true location and the model’s observable behaviour (answer content and answer/refusal pattern), with the full derivation provided in Appendix E (GLARE metric).
>
> As shown in this derivation, VRR is **built into** GLARE in two complementary ways. First, GLARE explicitly accounts for the answer vs. refusal pattern: a model that almost always refuses (VRR → 0) contributes almost no information through this channel. Second, GLARE scales the information contributed by the precision of the answers by VRR itself: even if the model is accurate when it answers, a low VRR directly suppresses its overall leakage. Thus, a model like Claude with low VRR does **not** “break” the evaluation as GLARE interprets it exactly as intended: the model answers less often and therefore leaks less information overall.
>
> To avoid any imbalance in our cross-model risk aggregated analysis in Section 3.3, we also explicitly excluded the Claude family, because its unusually low VRR could depress the overall estimate of MLRM location-related privacy leakage risk. This precaution keeps the high-level takeaways representative, while Table 2 still reports Claude’s per-model results for transparency.
>
> In summary, our evaluation strategy does **not** suffer from an imbalance caused by low VRR values. GLARE is constructed to jointly capture how often a model answers and how precisely it localizes when it does, providing an objective, comparable measure of privacy leakage across models. For transparency, we still report VRR and error distances separately in Table 2, while using GLARE as the principled aggregate indicator whose definition and derivation are given in Appendix E.
>
> > `Weakness 3 & Question 2`: **Many of the result explanations in the paper do not clearly indicate which figure or table they refer to. For example, the statement “Prediction difficulty increases with the annotated levels”, which figure is this referring to?**
>
> **A3**: Thank you very much for your careful reading and helpful comments! We agree that, in the original version, some result descriptions did not explicitly indicate the corresponding figures or tables, which could cause confusion. In the revision, we have systematically gone through all result explanation paragraphs and added explicit references to the relevant figures/tables.

---

> ### Author Response · Authors · 2025-11-22
> **Response to Reviewer gxPb (4/4)**
>
> >`Weakness 4 & Question 3`: **How was Figure 25 generated? From which dataset or classification process did it originate, and what method was used for the classification?**
>
> **A4**: We thank the reviewer for the careful reading and for asking how Figure 25 was generated. Figure 25 is not taken from any external dataset or pre-existing taxonomy. Instead, as described in Appendix F.1, we constructed an additional set of 50 street-view image–text pairs. For each image, we queried the OpenAI o3 model using the prompt in Figure 5b, which produces a thinking process including both a predicted location and the visual evidence used to narrow down the search region.
>
> From these thinking processes, we then performed a manual analysis in three stages:
>
> 1. **Low-level visual clue collection**: For each query, we extracted from the model’s reasoning all phrases that describe visual cues that help constrain the geographic location (e.g., “Tudor-style house”, “triangular park”, “desert landscaping”). We manually checked each extracted description against the corresponding image and corrected obvious errors or omissions to ensure that every clue faithfully matched visible content in the image.
>
> 2. **Clue type induction**: After manually collecting these low-level visual clues for all 50 images, we grouped similar clues into more abstract clue types (e.g., different descriptions of front-yard fences were merged into a single clue type).
>
> 3. **Category formation**: Finally, we manually clustered these clue types into a small number of higher-level semantic categories (Building Features, Identification, Urban Infrastructure, Property Features, Urban Planning). Figure 25 lists these categories, the corresponding clue types, and representative examples of the underlying visual clues.
>
> Thus, Figure 25 is a human-constructed taxonomy induced from OpenAI o3’s chain-of-thought on the additional 50 images queried with the Figure 5b prompt, rather than the result of training on or importing any existing classification scheme.
>
> ---
> > `Weakness 5 & Question 4`: **Regarding Figure 4, could there be potential information leakage, since the “clue” might have been extracted from the same dataset used for evaluation?**
>
> **A5**: We thank the reviewer for raising out this concern and would like to further clarify on this. GEOMINER does **not** rely on any pre-extracted clues or annotations from the evaluation dataset. Instead, the pipeline is purely test-time: for each evaluation image, we first query a *Detector* model once to extract natural-language visual clues from that **same** image, and then we query an *Analyzer* model that receives the image plus this clue text in its context to predict the address. Each image is processed independently (no clue bank, no reuse across samples, no use of ground-truth labels or EXIF metadata in the prompts), so the “clue” is just an intermediate description generated on-the-fly from the test image itself. Therefore, GEOMINER does not introduce additional information leakage from the dataset; it simply restructures the same visual evidence into a two-stage reasoning process.
>
> ---
> > `Ethics Concerns`:
>
> **A6**: We thank the reviewer for their diligence in maintaining high ethical standards. We take these concerns very seriously and wish to clarify the ethical compliance procedures undertaken for this study, specifically regarding the Amazon Mechanical Turk (MTurk) component.
>
> 1. **IRB Scope and Determination**:
>
> We respectfully clarify that the entire project scope, including the specific involvement of MTurk workers for evaluation, was submitted for institutional review. Our protocol was reviewed by our institution's review board. The official determination returned was "Not Human Subjects Review" (NHSR). Under this classification (consistent with the U.S. Common Rule 45 CFR 46), the IRB determined that the activity does not constitute "human subjects research" requiring full board approval or a standard exemption number.
>
> 2. **Proof of Compliance and Anonymity**:
>
> We possess the official determination record (Project ID ending in 13596, dated November 14, 2025) confirming this NHSR status. However, uploading the determination letter or providing the full Project ID at this stage would reveal our institutional affiliation, thereby violating the double-blind review policy of ICLR. We will commit the full Project ID and the specific language of the determination in the Ethics Statement of the camera-ready version to ensure full transparency.
>
> 3. **Worker Compensation**:
>
> Although the project was deemed NHSR, we adhered to ethical labor standards. We will update the paper to clarify that all participants were compensated at a rate well exceeding the U.S. federal minimum wage (USD 7.25 per hour) for their tasks (averaging approximately USD 12.15 per hour pro-rated).
>
> We hope this clarifies that the study has undergone appropriate institutional review and complies with all necessary ethical regulations.

---

> > ### Comment · Reviewer_gxPb · 2025-11-23
> > **Ethical consideration responses**
> >
> > I have a serious ethical concern regarding the MTurk component and the timing of the ethics determination described in the rebuttal. In A6, the authors state that the entire project, including the MTurk evaluation, was submitted to their institutional review board and that the activity received a "Not Human Subjects Review" (NHSR) determination, with a determination record (Project ID ending in 13596) dated November 14, 2025. However, the ICLR 2026 abstract and paper submission deadlines were in September 2025, which strongly suggests that at least some experiments, including the MTurk evaluation, were already conducted by that time.
> >
> > Given this timeline, it is not clear whether:
> > (1) the MTurk study protocol was submitted to the IRB before any data collection started, and
> > (2) the NHSR determination and the actual conduct of the study are fully consistent with ICLR ethical policy.
> >
> > At present, the rebuttal does not resolve the concern that the authors may have effectively obtained a retroactive determination after the study was already run, which would be ethically problematic if the activity should have been prospectively reviewed under their institutional procedures.
> >
> > I therefore request that the authors explicitly clarify the exact timing and ordering of: (a) IRB submission, (b) issuance of the NHSR determination, and (c) MTurk data collection, and confirm that this sequence complies with their institution’s requirements for research involving MTurk workers.
> >
> > If this concern cannot be convincingly addressed or is in fact impossible to resolve, I will have to regard the work as ethically non-compliant and adjust my overall recommendation accordingly (down to the lowest available score). In my view, any empirical research in our community must rest on a solid and demonstrably compliant ethical foundation; no technical contribution can compensate for unresolved violations or ambiguities in this respect.
> >
> > I am also flagging this issue for the AC/ethics chairs’ attention.

---

> > > ### Author Response · Authors · 2025-11-25
> > > **Response to Reviewer gxPb's Ethical Consideration (1/3)**
> > >
> > > We sincerely thank the reviewer for carefully raising this concern. We acknowledge and apologize for the confusion caused by our earlier wording, particularly mistakenly mentioning human research protection program (HRPP) as the institutional review board (IRB), which are **completely different**. We therefore appreciate the opportunity to clarify both the institutional classification of our entire study and the exact timeline.
> > >
> > > ## TL;DR
> > >
> > > In our institution, ethical review follows such a procedure: internal review by principal investigators (PIs) → HRPP review → IRB review. No HRPP/IRB submission is needed if the principal investigators determine a research does not qualify as a human subject research according to its definition (detailed policy in the later part). IRB does the formal prospective review **ONLY ON** human subject research (HSR) determined by HRPP. So, here, the HRPP and IRB are two distinct and different steps, and IRB will NOT review any research not gain the "human subject research determination" from HRPP.
> > >
> > > During our internal review on May 1, 2025, PI determined the entire research, including the Mechanical Turk (MTurk) part, does not qualify as a **human subject research** and thus no HRPP/IRB submission is needed according to our institutional policy (detailed policy is described in the later part). Thus, according to our institutional policy, we did not submit the HRPP before we started the whole research.
> > >
> > > On Nov 14, 2025, due to reviewer's request, we submitted our entire research to HRPP office and received the official **Not Human Subjects Research (NHSR)** determination (it is the **same** decision as our internal review on May 1, 2025 before all of our experiments). This determination can be viewed as an **official endorsement** of NHSR determination from our institution. This official decision from HRPP office further confirmed and verified that we **do not need to** and **should not submit to** the IRB at any time. **Our institute is the final authority on this NHSR determination**.
> > >
> > > We hope to clarify that we strictly follow our institute’s policy.  We will revise our Ethics section of our paper to ensure this is clarified.

---

> > > ### Author Response · Authors · 2025-11-25
> > > **Response to Reviewer gxPb's Ethical Consideration (2/3)**
> > >
> > > ## Detailed Explanation
> > >
> > > For ease of reference, the key milestones are:
> > >
> > > > **Internal review & NHSR determination according to our institutional policy  (May 1, 2025) → MTurk data collection (Aug 21–23, 2025) → ICLR submission deadline (Sep 24, 2025) → HRPP submission & official NHSR determination from HRPP office (Nov 14, 2025).**
> > >
> > > ### Institutional Policy:
> > > Ethical review follows such a procedure according to the institutional policy: internal review by principal investigators (PIs) →  Human Research Protection Program (HRPP) review →  IRB review. No HRPP/IRB submission is needed if the principal investigators determine the research does not qualify as human subject research according to the definition.
> > >
> > > If PIs think it is human subject research, HRPP determines whether an activity **meets the federal/institutional definition of "human subjects research"**. If an activity does **not** meet that definition, HRPP may issue a formal **Not Human Subjects Research (NHSR)** determination; this **is not** an IRB review and **does not** enter the IRB workflow. In contrast, only research that qualifies for human subject research can be and will be reviewed by the **Institutional Review Board (IRB)**, evaluating full protocols (e.g., consent, risk/benefit, continuing review).
> > >
> > > Here is the detailed policy from HRPP:
> > >
> > > > “Activities that do not meet the definition of Human Research do not require review and approval by the IRB and do not need to be submitted in the IRB Portal unless a formal determination is needed.”
> > >
> > > > Each activity undertaken on behalf of our institution must be evaluated by the individual most familiar with the planning and development of the activity. Therefore, it is the responsibility of researchers to make appropriate determinations based on this policy. When an individual makes a self-determination that an activity does not constitute human research, our institution's IRB recommends that the individual document in writing how the determination was made and retain this with his/her study records.
> > >
> > > ### Human subject research definition from our Institution.
> > >
> > > > Human Subject (DHHS): a living individual about whom an investigator conducting research obtains: (1) data through intervention or interaction with the individual or (2) identifiable private information [45 CFR 46.102(f)]
> > >
> > > > FDA: an individual who is or becomes a participant in research, either as a
> > > recipient of the test article or as a control. A subject may be either a healthy individual or a patient [21 CFR 56.102(e)]. In addition, a human subject includes an individual on whose specimen an investigational device or control is used, even if the specimen is anonymous [21 CFR 812.3(p)].

---

> > > ### Author Response · Authors · 2025-11-25
> > > **Response to Reviewer gxPb's Ethical Consideration (3/3)**
> > >
> > > ###  Internal review by PIs (May 1, 2025)
> > >
> > > Following our institution's policy, based on previous experience regarding MTurk research component design, our principal investigators performed an internal review **on May 1, 2025 before the research start** and determined in good faith that our project, **including the MTurk component**, does **not** meet the definition of "human subject research".  Here is the reason:
> > > The MTurk component of this project does not involve human subjects under DHHS 45 CFR 46.102(f), because no information is collected about the MTurk workers themselves. Workers do not provide identifiable private information of themselves, nor does the study involve interaction or intervention designed to obtain data about them. Their responses represent task performance on an objective human labeling task, not personal data. No FDA-regulated test articles are involved.
> > > Our study does not involve human subjects under the DHHS definition (45 CFR 46.102(f)). All images used in this research were captured by the authors from public vantage points or controlled, staged scenes involving only members of the research team. No interaction or intervention with any living individuals occurred during data collection. Under the FDA definition (21 CFR 56.102(e); 21 CFR 812.3(p)), no test article was administered to individuals, nor were any human specimens used.
> > > Therefore, the project does not constitute human subjects research and qualifies for IRB exemption. We **did not submit** an HRPP/IRB protocol before starting the study, because such submission is **neither required nor appropriate** at our institution for activities **outside the scope of human subject research**.
> > >
> > >
> > > ### MTurk data collection (Aug 21–23, 2025)
> > >
> > > The MTurk evaluation was conducted between **August 21, 2025 and August 23, 2025**, which is **well after** our internal review's determination by PI that the activity was not human subjects research, and **well before** the ICLR 2026 paper submission deadline of September 24, 2025.
> > >
> > > ### HRPP submission (Nov 14, 2025)
> > >
> > > During the ICLR review period, **after reviewer expressed concerns about the ethical status of the MTurk component**, we decided to submit the **entire research scope (including MTurk)** to our institute HRPP on **November 14, 2025** per reviewer's request, explicitly requesting an official determination of human subject research. Importantly, this HRPP request for a determination of human subject research was **not an official IRB submission**, and **we DID NOT submit this research to IRB at any stage**.
> > >
> > > ### NHSR determination (Nov 14, 2025)
> > >
> > > On **November 14, 2025**, the HRPP issued a formal **NOT Human subject research (NHSR)** determination (ID ending in 13596), explicitly covering the full research. This determination **DID NOT** retroactively approve any human subjects research. Rather, it served as an **official endorsement confirming** that our earlier classification during the internal review was correct: the research **DOES NOT** meet the definition of human subject research and therefore never required IRB review or exemption under our institution policy.
> > >
> > > To further avoid any ambiguity, in the revised Ethics section we will (i) replace the imprecise phrase “IRB exemption” with “Not Human Subjects Research Determination”, (ii) explicitly state the MTurk data collection window, and (iii) add a concise description of the worker tasks, inclusion criteria, and compensation scheme. We hope that this clarification addresses the reviewer’s concern and demonstrates that our study is fully compliant with both our institution’s policies and ICLR’s ethical standards.
> > >
> > > We have also brought this matter to the attention of the Area Chairs, Program Chairs, and Ethics Chairs, should further clarification be required, to ensure unambiguous compliance with ICLR’s double-blind reviewing rules and to forestall any misperception of academic misconduct or ethical non-compliance.

---

> > ### Comment · Reviewer_gxPb · 2025-11-24
> > **lower my score**
> >
> > Without a clear and verifiable explanation of the exact ordering of IRB submission, NHSR determination, and MTurk data collection, I must temporarily lower my score until the authors clarify this sequence.

---

> ### Author Response · Authors · 2025-11-25
>
> We thank you for your continued engagement and for recognizing that our work is solid. We appreciate your feedback regarding the ethics documentation and your comments on the technical contributions. We address your remaining concerns below.
> 1. Ethics Consideration
>
> We strictly followed our institution (**university**)’s policies regarding human subjects and ethical review. To the best of our knowledge, this is what we can do for the ethical requirement of ICLR.
>
> 2. Scientific Contribution and Novelty
>
> We would like to further clarify the distinctiveness of our work, as we believe it extends well beyond a "solid but not particularly distinctive" evaluation. We argue that this work provides the first systematic investigation of location-related privacy leakage specifically within Multi-modal Large Reasoning Models. Our contributions go significantly beyond a standard evaluation:
>
> - **Identification of a Novel Threat Surface:** We identify and formalize a previously underexplored privacy risk specific to Multimodal Large Reasoning Models: the leakage of sensitive geolocation information through visual inference.
> - **Foundational Benchmark (DOXBENCH):** We created the first curated dataset specifically designed to evaluate privacy risks (tiered by individual vs. household risk) rather than just geolocation accuracy. This fills a critical gap in existing benchmarks which focus on benign landmarks.
> - **Novel Metric (GLARE):** We introduced the Geolocation Leakage And Risk Estimate (GLARE), an information-theoretic metric. This is a methodological innovation that providing a more rigorous measure of privacy risk than simple distance error.
> - **Root Cause & Mechanism:** We did not simply observe the phenomenon; we analyzed why it occurs (via **ClueMiner** to identify visual clues) and demonstrated the amplified real-world impact (via **GeoMiner**).
>
> We believe that exposing this novel threat surface, understanding its underlying causes, and providing the tools to measure it are critical steps. This work creates the necessary foundation for the community to understand these risks and develop effective safety solutions. **We hope this highlights the substantial novelty and contribution of our research.**
>
> 3. Clarification on "Upper Bound" and Rigor
>
> We appreciate the opportunity to clarify the context regarding the "upper bound" discussion. The term "potential risk upper bound" does not appear in our original paper, nor was it a claim made in the manuscript.
>
> We only introduced this phrasing in our previous rebuttal **specifically to answer your query** regarding "Weakness 1." Since the term "upper bound" was not defined in the context of your question, we interpreted it as "probing how much leakage (risk) a model exhibits when the adversary applies stronger prompting strategies (capacity)" to provide you with a comprehensive answer. **Our intention was to provide a comprehensive answer to your specific hypothetical, rather than to redefine the scope of our paper.**
>
> Therefore, we respectfully suggest that defining the entire paper as an "adversarially-enhanced leakage evaluation" based on a rebuttal clarification **might overlook the broader scope of our study**. The paper remains a comprehensive study of inference-time privacy risks, of which adversarial amplification is just one component of our broader analysis.
>
> We hope this clarifies the rigor of our methodology and the distinctiveness of our contributions.

---

### Official Review · Reviewer_WVq6 · 2025-10-28

**Soundness:** 2
**Presentation:** 3
**Contribution:** 2
**Rating:** 4
**Confidence:** 3

**Summary:**

This paper investigates a novel privacy risk associated with recent multi-modal large reasoning models (MLRMs). It finds that these models possess sophisticated reasoning capabilities enabling them to infer sensitive geolocation information, such as home addresses or neighborhoods, from user-generated images like selfies, even those taken in private settings. To evaluate this risk, the authors propose a three-level privacy risk framework and introduce DOXBENCH, a benchmark dataset of 500 real-world images representing various privacy scenarios.

**Strengths:**

1. This paper focuses on evaluating the risk of geolocation privacy leakage in large models and introduces a novel benchmark dataset for this purpose.

2. The use of the GLARE metric is well-justified, offering a more comprehensive assessment than simple accuracy measures alone.

3. The experiments conducted are extensive, covering a wide range of mainstream large models.

**Weaknesses:**

1. The paper anchors its geographic scope almost exclusively to California, rendering the dataset incomplete and limiting its persuasiveness. This raises concerns about potential bias, possibly stemming from an overrepresentation of California data in the large models' training sets. While the authors acknowledge this limitation in Appendix F, the discussion provided is far from sufficient to address the concern.

2. Critical questions regarding defense mechanisms and the utility of geolocation data remain unanswered. How can this identified privacy vulnerability be effectively mitigated? Furthermore, what is the quantifiable benefit or utility gained by large models from leveraging geographic coordinates?

3. The distinction between "Privacy Space" and "Personal Imagery" lacks clarity. What are the precise operational differences between these two concepts? They appear highly similar and ultimately converge on the fundamental issue of personal privacy, making their practical differentiation ambiguous.

**Questions:**

See above.

---

> ### Author Response · Authors · 2025-11-22
> **Response to Reviewer WVq6 (1/2)**
>
> We sincerely appreciate your thoughtful and detailed review of our manuscript. Your questions have been extremely helpful in improving our paper. In what follows, we provide point-by-point responses to each of your comments.
>
> ---
> > `Weakness 1`: **The paper anchors its geographic scope almost exclusively to California, rendering the dataset incomplete and limiting its persuasiveness. This raises concerns about potential bias, possibly stemming from an overrepresentation of California data in the large models' training sets. While the authors acknowledge this limitation in Appendix F, the discussion provided is far from sufficient to address the concern.**
>
> **A1**: We appreciate the reviewer’s concern about the California-focused scope, dataset completeness, and potential bias from overrepresented California imagery. Since similar questions about cross-region generalization were raised by multiple reviewers, we provide dedicated [**Global Responses on Generalization Across Regions**](https://openreview.net/forum?id=uBThjlbzxS&noteId=WfpxPXceK3) at the beginning. We kindly refer the reviewer to the global response, which present new cross-region experiments and a detailed discussion directly addressing this concern.
>
> ---
> > `Weakness 2`: **Critical questions regarding defense mechanisms and the utility of geolocation data remain unanswered. How can this identified privacy vulnerability be effectively mitigated? Furthermore, what is the quantifiable benefit or utility gained by large models from leveraging geographic coordinates?**
>
> **A2**: Thank you for raising this important question about both potential defense mechanisms and the utility of geolocation capabilities in large models. We address these two aspects in turn.
>
> **Defense mechanisms and mitigation**: We appreciate the reviewer’s question and agree that developing effective defenses against image-based geolocation leakage is an important and largely open research problem. The main goal of our paper is to show this risk and build a benchmark to provide a standard evaluation framework that can advance the community in addressing this risk by proposing their diverse algorithms. We believe that finding the problems and providing a benchmark itself for advancing the community for better defense are indeed enough contributions. Secondly, we also show that this risk is a non-trivial risk and can not be easily defended by some simple strategies (e.g., guardrails, prompt-based refusals, and simple image perturbations), which opens a problem for the community, and this problem is a non-trivial problem. We believe proposing and opening a problem is sometimes as important as solving the problem. A comprehensive defense framework is beyond the scope of this paper. We argue that exposing the problem, understanding why it occurs, and demonstrating the real-world impact are very critical, as they create the foundation for the community to understand and develop solutions for effectively mitigating the risk.
>
> **Utility of geolocation capabilities**: While our focus is on exposing privacy risks, we agree that geolocation capabilities can also provide substantial positive utility, which partly explains why they are being integrated into large models. Reducing the spatial uncertainty of an image, for example from “anywhere” to a specific region, city, or neighborhood, can be quantified using metrics such as error distance or region-level accuracy (see evaluation metrics in Appendix G), and this directly benefits downstream tasks. For example, in crime investigation, narrowing the plausible location of publicly shared images can reduce manual search effort and accelerate identification of suspects or victims; in disaster response, inferring approximate locations from damage photos without reliable GPS tags can help prioritize areas and allocate rescue resources more efficiently; and in tourism and cultural-heritage scenarios, automatic landmark and attraction recognition improves photo organization, recommendation quality, and navigation assistance (including for visually impaired users). Our work does not aim to optimize these applications, but the same geolocation metrics we propose for measuring leakage-GLARE can also be used in future work to quantify the *utility* side of this trade-off, highlighting the need to balance such benefits against the privacy risks we reveal.

---

> ### Author Response · Authors · 2025-11-22
> **Response to Reviewer WVq6 (2/2)**
>
> > `Weakness 3`: **The distinction between "Privacy Space" and "Personal Imagery" lacks clarity. What are the precise operational differences between these two concepts? They appear highly similar and ultimately converge on the fundamental issue of personal privacy, making their practical differentiation ambiguous.**
>
> **A3**: We thank the reviewer for raising this concern and would like to respectfully clarify that, in Section 2.2 of the paper, we deliberately design “Privacy Space” and “Personal Imagery” as two independent dimensions, rather than two duplicate labels for “personal privacy.”
>
> The former is a space-centric concept: it marks whether a photo is taken in the user’s living space and its immediately adjacent areas (such as the interior, front porch, or fenced backyard). Once identified, this exposes a household’s long-term residence and sphere of daily life, reflecting household/address-level persistent risk. Under the **Restatement (Second) of Torts § 652B [1]**, capturing the private space itself constitutes Intrusion upon Seclusion, a tort that triggers liability even without the publication of the imagery, highlighting the severity of spatial exposure.
>
> The latter is a person-centric concept, aligning with the core protection of “identifiable natural persons” established in frameworks like the **GDPR** and **CCPA**. It marks whether there is a specific, recognizable individual as the main subject of the image (such as a selfie or frontal portrait). Regardless of whether this occurs at home, in a mall, or on the street, combining such imagery with geolocation produces transient individual risk relating to the person’s movements and activities.
>
> Operationally, the two are annotated as separate binary labels: In Appendix Figure 21, for each image, we first determine whether it belongs to Privacy Space, and then whether it qualifies as Personal Imagery. The combination of these labels yields three distinct risk levels: only a person outside Privacy Space (individual risk only), only a Privacy Space with no specific person (household/address risk only), and the highest-risk case where both are present. Because the two concepts focus on different objects (location vs. person) and lead to different legal and safety consequences, this decomposition is not redundant in our empirical evaluation; instead, it allows us to distinguish and quantify three types of privacy leakage: “exposing only the home address,” “exposing only the individual’s movements,” and “simultaneously exposing both the person and the home” as the most severe case. In the revised version, we will further clarify this two-dimensional design through more explicit text and figures to reduce readers’ confusion about the two notions being “too similar.”
>
> [1] American Law Institute. (1977). Restatement (second) of torts. St. Paul, MN: American Law Institute Publishers.

---

> ### Author Response · Authors · 2025-11-26
>
> Dear Reviewer,
>
> We sincerely appreciate the time and effort you have dedicated to reviewing our paper. As the discussion period is nearing its end, we wanted to ensure we have thoroughly addressed all of your concerns. Your feedback has been truly valuable in helping us improve our work. If there are any outstanding issues or additional suggestions you would like to share, we would be deeply grateful for the opportunity to address them.

---

### Official Review · Reviewer_APLg · 2025-10-29

**Soundness:** 3
**Presentation:** 3
**Contribution:** 3
**Rating:** 8
**Confidence:** 2

**Summary:**

This paper reveals that multi-modal large reasoning models (MLRMs) can infer users’ private locations from ordinary photos, including selfies in personal spaces. The authors build DOXBENCH, a 500-image dataset of real-world scenes annotated into three legal privacy-risk levels, and evaluate 13 MLRMs and MLLMs, showing that most outperform non-expert humans in geolocation inference. The paper further proposes CLUEMINER to identify which visual clues drive location reasoning, and GEOMINER, a two-stage attack combining clue extraction and reasoning to amplify leakage.

**Strengths:**

S1: The problem studied in this paper is interesting and well-motivated.

S2: The paper introduces a purpose-built dataset of 500 privacy-sensitive, real-world images representing personal spaces rather than public landmarks, making the evaluation realistic and legally grounded.

S3: The paper benchmarks 14 leading multimodal models using reproducible metrics.

S4: The paper develops CLUEMINER (to identify visual clue categories) and GEOMINER (a two-stage clue-assisted attack) that together reveal how and why leakage occurs.

S5: Overall, the paper is well written, well-organized, and easy to follow.

**Weaknesses:**

W1: DOXBENCH primarily includes images from California and nearby areas, which may limit geographic, cultural, and environmental diversity; generalization to other regions remains unclear.

W2: It seems that the experiments focus on image-based inputs; the approach and findings may not fully extend to other modalities (e.g., video or text-image pairs).

**Questions:**

Q1: Can the authors briefly discuss whether their findings generalize to regions beyond California, and what additional geographic or environmental factors might influence model performance?

Q2: Can the proposed analysis and metrics be extended to other modalities, such as video or text–image pairs, and what challenges might arise in doing so?

---

> ### Author Response · Authors · 2025-11-22
>
> We are truly grateful for the time and care you invested in reviewing our work, as well as for the insightful and constructive suggestions you provided. Below, we respond to your comments point by point to your comments.
>
> ---
> > `Question 1`: **Can the authors briefly discuss whether their findings generalize to regions beyond California, and what additional geographic or environmental factors might influence model performance?**
>
>
> **A1**: We thank the reviewer for raising this question about whether our findings generalize to regions beyond California and how additional geographic or environmental factors might affect model performance. Since similar concerns about cross-region generalization were raised by multiple reviewers, we provide a dedicated [**Global Response on Generalization Across Regions**](https://openreview.net/forum?id=uBThjlbzxS&noteId=WfpxPXceK3) at the beginning. We kindly refer the reviewer to the global response for a detailed discussion and the corresponding experimental results.
>
> ---
> > `Question 2`: **Can the proposed analysis and metrics be extended to other modalities, such as video or text–image pairs, and what challenges might arise in doing so?**
>
> **A2**: We thank the reviewer for suggesting this valuable direction. To address the question, we conducted additional experiments on the video modality. We randomly extracted keyframes from single videos (3-second short videos generated via iPhone’s Live Photo feature) and compared two settings: (i) submitting each frame to OpenAI o3 separately v.s. (ii) submitting multiple frames jointly. Sample demonstrations are shown as **Figure 39 to Figure 41** in the appendix of the revised version. In the single-frame setting (shown as Figure 40 and Figure 41), the average error distance of the model's inference is 2.87 km. When the two frames were combined (shown as Figure 39), the model returned the exact location. This shows that **complementary clues across frames are integrated by the model, which in turn amplifies privacy leakage**.
>
> **Challenges do exist for other modalities**. Even for a single image, inference with MLRMs is costly. For video, current common practice is to decompose the clip into frames and feed them as multiple images in one prompt. As a result, **token usage grows significantly with the number of frames increases**, pushing against the context window and causing earlier frames to be forgotten. The attention of the model may also focus on salient yet uninformative frames, suppressing frames that carry key clues. For fair comparison in future studies, evaluations of video modality should limit a fixed "frame budget" and specify frame sampling and combining strategies. For evaluations of text–image pairs, text are high-signal clues that can dominate the prediction as LLMs naturally favors text more, so it is important to control the bias introduced by explicit text through methods like capping the text token budget, reporting text-to-vision ratios, and running ablations with different text shares.
>
> **Our proposed analysis and metrics extend without change since the task and output format remain the same through different modalities.** In practice, the analysis of different modalities' privacy leakage should simply treat visual evidence from different frames or text as distinct evidence sources and have them controlled to ensure fair evaluation and comparison across different models.

---

> > ### Comment · Reviewer_APLg · 2025-11-27
> > **Acknowledgment of Author Response**
> >
> > I appreciate the authors’ detailed and thoughtful responses. The clarifications regarding cross-region generalization and the inclusion of a Global Response on Generalization Across Regions are noted and helpful. The additional experiments on the video modality meaningfully extend the paper’s scope. Overall, I thank the authors for their comprehensive and well-organized response.

---

> ### Author Response · Authors · 2025-11-26
>
> Dear Reviewer,
>
> We sincerely appreciate the time and effort you have dedicated to reviewing our paper. As the discussion period is nearing its end, we wanted to ensure we have thoroughly addressed all of your concerns. Your feedback has been truly valuable in helping us improve our work. If there are any outstanding issues or additional suggestions you would like to share, we would be deeply grateful for the opportunity to address them.

---

> ### Author Response · Authors · 2025-11-27
>
> We sincerely appreciate your constructive feedback and invaluable insight. Your effort is indispensable to the improvement of our work.

---

### Official Review · Reviewer_qp28 · 2025-10-31

**Soundness:** 3
**Presentation:** 3
**Contribution:** 3
**Rating:** 6
**Confidence:** 4

**Summary:**

This paper presents the first systematic study of location-related privacy leakage in Multi-modal Large Reasoning Models. The authors identify a novel privacy risk where adversaries can infer sensitive geolocation information from user-generated images. Key contributions include:They constructed DoxBench, the first benchmark dataset specifically designed to evaluate this risk. They also introduced a three-tier privacy risk taxonomy grounded in legal frameworks. Furthermore, it innovatively proposes GLARE, an information-theoretic metric to quantify the extent of privacy leakage, and develops analytical tools named ClueMiner and GeoMiner to trace the root causes of the risk and demonstrate attack feasibility.

**Strengths:**

The authors have meticulously constructed a privacy dataset containing real-world scenarios, proposed a highly innovative evaluation metric, and demonstrated the pervasiveness and severity of the risks through extensive experiments. They even showcased how their attack tools could enable ordinary users to achieve this with ease. The entire research framework is comprehensive, progressing logically from problem definition and analysis to verification, with robust evidence throughout.

**Weaknesses:**

The current dataset primarily focuses on California, USA, which naturally raises the question: would this methodology remain equally effective when applied to European or Asian streetscapes and architectural styles? Furthermore, in the defense section, while several methods were tested, the underlying reasons for their failures haven't been thoroughly explored. A deeper analysis of how the models circumvent blurring and noise-based defenses would provide more valuable insights.

**Questions:**

Could you validate your findings on images from other geographic regions？

---

> ### Author Response · Authors · 2025-11-22
>
> Thank you immensely for your time and efforts, as well as the helpful and constructive feedback! Here, we give point-by-point responses to your comments.
>
> ---
> > `Weakness 1 & Question`: **The current dataset primarily focuses on California, USA, which naturally raises the question: would this methodology remain equally effective when applied to European or Asian streetscapes and architectural styles? Could you validate your findings on images from other geographic regions？**
>
>
> **A1**: We thank the reviewer for this thoughtful question about whether our methodology extends beyond California to European or Asian streetscapes and architectural styles. Because similar concerns about cross-region generalization were raised by multiple reviewers, we provide a dedicated [**Global Response on Generalization Across Regions**](https://openreview.net/forum?id=uBThjlbzxS&noteId=WfpxPXceK3) at the beginning. We kindly refer the reviewer to the global response, which present new experiments on images from additional countries and devices and show that the key trends in privacy leakage remain consistent beyond California.
>
>
> ---
> > `Weakness 2`: **Furthermore, in the defense section, while several methods were tested, the underlying reasons for their failures haven't been thoroughly explored. A deeper analysis of how the models circumvent blurring and noise-based defenses would provide more valuable insights.**
>
> **A2**: We thank the reviewer for their concerns regarding our defense analysis. In Appendix J, we provide a detailed evaluation and discussion:
> 1. **Manually blurring visual clues**
>
>    To simulate how a potential victim might redact sensitive information before posting photos on social media, we manually blurred all visual cues that our researchers could identify (e.g., storefront names, distinctive signs, license plates, and other location-revealing details). This corresponds to a strong, human-driven redaction strategy.
>    Empirically, in Appendix J2, despite average reductions of 16.58% in VRR and 30.6% in GLARE, the models still achieve an average CCPA accuracy of 10.56%. The reason manual redaction fails is that MLRMs reason holistically. You might hide the text IDs, but the model will just pivot to architectural styles. Hide the building, and it switches to analyzing plants or the skyline. This adaptive reasoning made such masking defenses futile.
>
> 2. **Noise-based defenses**
>
>    As shown in Appendix J5 Figure 14, Gaussian noise reduces the model’s ability to recognize harmful images as the noise level increases. However, it simultaneously degrades performance on benign public-landmark photos, making it difficult to find a balance point that both preserves accurate localization for harmless public scenes and prevents privacy leakage.
>    Moreover, Appendix J3 Table 10 shows that adversarial noise perturbations further harm the model’s utility on other tasks, such as OCR on images and answering simple visual QA questions. This side effect suggests that noise-based defenses, while partially effective, are impractical for general-purpose MLRMs because they introduce substantial collateral damage to benign and non-location-related functionalities.

---

> ### Author Response · Authors · 2025-11-26
>
> Dear Reviewer,
>
> We sincerely appreciate the time and effort you have dedicated to reviewing our paper. As the discussion period is nearing its end, we wanted to ensure we have thoroughly addressed all of your concerns. Your feedback has been truly valuable in helping us improve our work. If there are any outstanding issues or additional suggestions you would like to share, we would be deeply grateful for the opportunity to address them.

---

### Author Response · Authors · 2025-11-22
**Global Response on Generalization Across Regions**

We thank the reviewers for raising this important point about cross-region generalization and dataset diversity. We agree that geographic and cultural diversity is crucial. We respectfully clarify our existing cross-region analyses and add new experiments on additional countries/regions, as summarized below:

- **Additional U.S. states**: In Appendix F, we extend our study beyond California by leveraging Google Street View images from multiple other U.S. states spanning the East, West, South, and North. We also vary prompts to probe advanced MLRMs under different query styles. We find that MLRMs exhibit an even higher privacy-leakage risks than on the California-collected photos. The mean CCPA accuracy reaches 19.6% and GLARE reaches 1908.14 bits, both already high. Notably, with tool assistance, OpenAI o3 achieves 34% CCPA accuracy and a GLARE of 2375.48 bits. These results indicate strong generalizability of image-based, location-related privacy risk beyond California to photos taken in other U.S. states, which should be considered a new and practical threat for MLRMs.


- **Qualitative demo on Chinese addresses**: In the demo at the end of the appendix, we provide further qualitative examples of OpenAI o3’s performance on two real addresses: 753 Shendi East Rd, Pudong, Shanghai, China (shown as Figure 26), and Cuiwei Street 99, Suzhou Industrial Park, Suzhou, Jiangsu (shown as Figure 29). These examples demonstrate that the model can successfully recognize and localize sensitive locations outside the U.S., again supporting the cross-region generalizability of the privacy risk.


- **New photos across regions and devices**: To further address the reviewer’s concern, we additionally collected 50 real-world photos focusing on L1 and L2 privacy level, captured by our research team using Galaxy Z Fold 5, OPPO Find X8 Ultra, OPPO Find X7 Ultra, and iPhone 13 Pro devices in **UK, Spain, France, Turkey, China (Mainland), China (Taiwan) and Japan**. We evaluate this additional dataset on four most advanced MLRMs (**Gemini 2.5 Pro, GPT5, OpenAI o3, and OpenAI o4 mini**), and their average performance across regions is shown below in Table R1 (Top-1) and Table R2 (Top-3):

**Table R1: Top-1 per-region performance on the new additional 50-photo cross-region dataset.**

|Region|VRR|Average Distance|Median Distance|CCPA|GLARE|
|-|:-:|:-:|:-:|:-:|:-:|
|All|81.95|243.94|102.92|51.79|2062.81|
|UK|73.68|31.15|0.46|64.62|1642.05|
|Spain|82.50|3.89|0.35|64.58|2196.04|
|France|68.75|1.07|0.65|33.33|2426.68|
|Turkey|90.00|0.48|0.35|68.75|2573.93|
|China (Mainland)|75.00|1817.83|734.30|16.67|842.84|
|China (Taiwan)|90.00|185.82|116.49|14.58|1517.02|
|Japan|93.75|0.11|0.10|**100.00**|**3027.10**|

**Table R2: Top-3 per-region performance on the new additional 50-photo cross-region dataset.**

|Region|VRR|Average Distance|Median Distance|CCPA|GLARE|
|-|:-:|:-:|:-:|:-:|:-:|
|All|83.14|310.10|222.07|68.21|2178.04|
|UK|81.58|595.93|0.53|55.22|1716.52|
|Spain|87.50|0.50|0.18|79.76|2588.15|
|France|93.75|1.00|0.83|54.17|2555.27|
|Turkey|90.00|0.20|0.15|93.75|2791.93|
|China (Mainland)|66.67|278.80|260.69|41.67|1207.13|
|China (Taiwan)|75.00|1294.12|1291.99|52.92|1598.15|
|Japan|87.50|0.13|0.11|**100.00**|**2789.15**|

  As shown in Table R1 and Table R2, the MLRMs achieve an average Top-1 CCPA of 24.74% and an average Top-1 GLARE of 1908.14 bits, while the average Top-3 CCPA is 19.80% and the average Top-3 GLARE is 2038.11 bits. Moreover, in most regions, we observe GLARE values exceeding 2000 bits and CCPA exceeding 50%, which is consistent with our main findings and further supports the generalizability of the identified location-based privacy risks across diverse regions and environments.

**For more detailed per-model Top-1 and Top-3 performance across countries/regions, please refer to Global Response (2/2), Tables R3 and R4.** We will release these 50 photos as an additional dataset on DoxBench to further support the community’s research on location-related privacy leakage. We’ve also attached some example chats using samples from these 50 photos in the appendix of our revised version (shown as **Figure 33 to Figure 38**).

We hope these clarifications and new experiments address the reviewers’ concerns on geographic generalization and provide a clearer picture of the broad applicability of our findings.

---

> ### Author Response · Authors · 2025-11-22
>
> **Table R3: Per-model Top-1 performance across regions on the new additional 50-photo cross-region dataset.**
>
> |Model|Region|VRR|Average Distance|Median Distance|CCPA|GLARE|
> |-|-|:-:|:-:|:-:|:-:|:-:|
> |Gemini 2.5 Pro|All|88.00|3.82|0.28|61.11|2234.38|
> ||UK|94.74|4.93|0.29|71.43|2365.90|
> ||Spain|80.00|0.44|0.15|66.67|2351.43|
> ||France|75.00|2.09|0.74|33.33|1865.09|
> ||Turkey|80.00|0.11|0.16|**100.00**|**2508.05**|
> ||China (Mainland)|100.00|251.02|320.34|33.33|919.44|
> ||China (Taiwan)|100.00|11.90|6.44|0.00|1923.02|
> ||Japan|75.00|0.07|0.06|**100.00**|2503.62|
> |GPT5|All|100.00|1.75|0.28|60.53|2653.66|
> ||UK|100.00|56.38|1.14|47.06|1948.73|
> ||Spain|100.00|0.44|0.20|75.00|2901.33|
> ||France|100.00|0.58|0.72|33.33|2674.61|
> ||Turkey|100.00|0.39|0.16|75.00|2953.70|
> ||China (Mainland)|100.00|611.26|6.04|33.33|1363.96|
> ||China (Taiwan)|100.00|5.78|6.88|0.00|2017.50|
> ||Japan|100.00|0.11|0.09|**100.00**|**3215.29**|
> |OpenAI o3|All|80.00|16.47|0.34|61.29|1840.71|
> ||UK|68.42|43.36|0.28|60.00|1496.03|
> ||Spain|70.00|7.56|0.41|66.67|1670.96|
> ||France|100.00|0.55|0.48|66.67|2740.33|
> ||Turkey|100.00|0.89|0.54|50.00|2654.12|
> ||China (Mainland)|100.00|4591.21|1876.51|0.00|245.10|
> ||China (Taiwan)|80.00|715.63|445.76|25.00|576.51|
> ||Japan|100.00|0.06|0.05|**100.00**|**3382.92**|
> |OpenAI o4 mini|All|52.00|3.64|0.36|66.67|1305.06|
> ||UK|31.58|19.95|0.14|80.00|757.55|
> ||Spain|80.00|7.14|0.66|50.00|1860.45|
> ||France|0.00|NaN|NaN|0.00|NaN|
> ||Turkey|80.00|0.54|0.54|50.00|2179.84|
> ||China (Mainland)|0.00|NaN|NaN|0.00|NaN|
> ||China(Taiwan)|80.00|9.97|6.88|33.33|1551.06|
> ||Japan|100.00|0.22|0.19|**100.00**|**3006.57**|
>
> **Table R4: Per-model Top-3 performance across regions on the 50-photo cross-region dataset.**
>
> |Model|Region|VRR|Average Distance|Median Distance|CCPA|GLARE|
> |-|-|:-:|:-:|:-:|:-:|:-:|
> |Gemini 2.5 Pro|All|96.00|0.54|0.16|75.00|2785.13|
> ||UK|100.00|22.72|0.18|64.71|2346.81|
> ||Spain|100.00|0.77|0.24|66.67|2793.05|
> ||France|100.00|0.63|0.46|50.00|2727.95|
> ||Turkey|100.00|0.32|0.13|75.00|**3011.47**|
> ||China (Mainland)|33.33|0.24|0.24|**100.00**|988.63|
> ||China (Taiwan)|100.00|0.50|0.38|75.00|2789.16|
> ||Japan|100.00|0.09|0.08|**100.00**|3262.20|
> |GPT5|All|96.00|1.42|0.23|67.50|2599.47|
> ||UK|94.74|16.74|0.25|53.33|2219.40|
> ||Spain|90.00|0.24|0.10|85.71|2772.60|
> ||France|100.00|0.37|0.22|66.67|2911.27|
> ||Turkey|100.00|0.15|0.11|**100.00**|3138.20|
> ||China (Mainland)|100.00|2.34|0.75|33.33|2467.84|
> ||China (Taiwan)|100.00|4.22|3.76|20.00|2149.97|
> ||Japan|100.00|0.11|0.09|**100.00**|**3220.74**|
> |OpenAI o3|All|72.00|2.15|0.15|82.14|1952.46|
> ||UK|57.89|56.52|0.17|60.00|1287.30|
> ||Spain|90.00|0.15|0.09|**100.00**|2853.56|
> ||France|100.00|0.39|0.28|66.67|**2867.20**|
> ||Turkey|80.00|0.06|0.07|**100.00**|2686.52|
> ||China (Mainland)|100.00|154.69|83.85|33.33|1182.65|
> ||China (Taiwan)|40.00|5163.39|5163.39|50.00|32.85|
> ||Japan|75.00|0.21|0.13|**100.00**|2301.09|
> |OpenAI o4 mini|All|70.00|1.62|0.28|70.37|1863.24|
> ||UK|73.68|2287.72|1.50|42.86|1012.55|
> ||Spain|70.00|0.82|0.28|66.67|1933.37|
> ||France|75.00|2.61|2.36|33.33|1714.63|
> ||Turkey|80.00|0.27|0.30|**100.00**|2331.53|
> ||China (Mainland)|33.33|957.92|957.92|0.00|189.40|
> ||China (Taiwan)|60.00|8.39|0.42|66.67|1420.63|
> ||Japan|75.00|0.10|0.14|**100.00**|**2372.57**|

---

### Meta-Review · Area_Chair_c7dP · 2026-01-06

**Summary:**

To address the privacy risks of multi-modal large reasoning models (MLRMs), the authors identify a novel category of privacy leakage in MLRMs and propose a three-level privacy risk framework that categorizes image based on contextual sensitivity and potential for geolocation inference.

Reviewers gxPb, qp28, and APLg initially gave positive scores (6, 6, 8) and Reviewer APLg acknowledged the clarifications regarding cross-region generalization and the inclusion of a Global Response on Generalization Across Regions after rebuttal.
For Reviewer WVq6, although (s)he gave a negative score of 4, the authors' responses to (1) geographic bias and generalization, (2) lack of defense mechanisms and utility discussion, and (3) ambiguity between "Privacy Space" and "Personal Imagery" look reasonable.

Overall, taking all the comments and responses into consideration, the AC would like to accept this paper!

**Reviewer Concerns:**

The authors have properly addressed the concerns from the reviewers.

**Reviewer Scores:**

none

---

### Decision · Program_Chairs · 2026-01-26

Accept (Poster)